# In vivo mapping of sodium homeostasis disturbances in individual ALS patients: A brain $^{23}$Na MRI study

**Aude-Marie Grapperon**[1,2,3]*, **Mohamed Mounir El Mendili**[1,2], **Adil Maarouf**[1,2,4], **Jean-Philippe Ranjeva**[1,2], **Maxime Guye**[1,2], **Annie Verschueren**[1,2,3], **Shahram Attarian**[3,5], **Wafaa Zaaraoui**[1,2]

**1** CNRS, CRMBM, Aix Marseille Univ, Marseille, France, **2** APHM, Hôpital de la Timone, CEMEREM, Marseille, France, **3** APHM, Hôpital de la Timone, Referral Center for Neuromuscular Diseases and ALS, ERN EURO-NMD, Marseille, France, **4** APHM, Hôpital de la Timone, Pôle de Neurosciences Cliniques, Service de Neurologie, Marseille, France, **5** INSERM, Marseille Medical Genetic Center, Aix Marseille Univ, Marseille, France

* aude-marie.grapperon@ap-hm.fr

**Data Availability Statement:** All relevant data are within the manuscript and its Supporting information files. Due to legal and ethical constraints surrounding potentially identifying

## Abstract

### Objective

Amyotrophic lateral sclerosis (ALS) is a neurodegenerative disease characterized by significant heterogeneity among patients. $^{23}$Na MRI maps abnormal sodium homeostasis that reflects metabolic alterations and energetic failure contributing to the neurodegenerative process. In this study, we investigated disease severity at the individual level in ALS patients using brain $^{23}$Na MRI.

### Methods

$^{1}$H and $^{23}$Na brain MRI were collected prospectively from 28 ALS patients. Individual map of abnormal total sodium concentration (TSC) was computed using voxel-based statistical mapping for each patient compared to a local database of 62 healthy controls. Clinical data included the revised ALS functional rating scale (ALSFRS-R), ALSFRS-R slope, ALSFRS-R at 6-month and survival time.

### Results

Individual maps quantifying voxels with TSC increase evidenced a high heterogeneity between patients consistent with clinical presentation. The main areas involved were the corticospinal tracts. Half of patients showed abnormal TSC increase within more than 1% of whole brain voxels. Patients with TSC increase had worse clinical severity: higher ALSFRS-R slope (p = 0.02), lower ALSFRS-R at 6-month (p = 0.04), and shorter survival (p = 0.04). ALS patients with limited TSC increase had slower progression of disability or predominant lower motor neuron phenotype or shorter disease duration.

information, MRI data cannot be shared publicly. Requests for access to the data will be evaluated on a case-by-case basis, subject to a data usage agreement (please email drci@ap-hm.fr).

**Funding:** This research was funded by Assistance Publique Hopitaux de Marseille (APHM, AORC Junior program), ARSLA (Association pour la Recherche sur la Sclérose Latérale Amyotrophique et autres maladies du motoneurone) and FRC (Fédération pour la Recherche sur le Cerveau). We certify that the funders had no role in study design, data collection and analysis, decision to publish, or preparation of the manuscript.

**Competing interests:** The authors have declared that no competing interests exist.

## Discussion

This study mapping sodium homeostasis disturbances at the individual level in ALS patients through $^{23}$Na MRI evidenced heterogeneity of TSC increase among patients associated with clinical presentation and disease severity. These findings suggest that TSC increase detected at the individual level by $^{23}$Na MRI may be a useful marker of the clinical heterogeneity of ALS patients, a factor that is likely to greatly influence the results of therapeutic trials.

## 1. Introduction

Amyotrophic lateral sclerosis (ALS) is a neurodegenerative disease that leads to progressive motor deficit and ultimately death within few years. The disease is characterized by upper and lower motor neuron degeneration. Disease progression is variable among patients [1]. While the median survival time ranges from 20 to 48 months, 10 to 20% of patients have a survival longer than 10 years. A usual clinical criterion to assess patient's disability is the revised ALS functional rating scale (ALSFRS-R) [2]. The ALSFRS-R progression shows also high variability between patients demonstrating the heterogeneity of the disease that reflects the various and partially understood processes involved in ALS. Thus, there is an unmet need to identify non-invasive biomarkers to characterize the disease in patients at the individual level and to predict the progression of disability [3].

Several MRI techniques, such as diffusion tensor imaging (DTI) or cortical thickness measurements, have reported widespread structural damage in motor but also non-motor brain regions in ALS patients at the population level [4–16]. Brain damage revealed by MRI techniques in patients with ALS is well-documented but remains inconsistent due to the high heterogeneity of the disease. Recently, imaging techniques have helped to identify distinct disease-burden patterns, aiding in patient stratification [17–20]. However, the majority of studies focus on group comparisons, typically between ALS patients and healthy controls, which has led to a growing gap between group-level findings in ALS neuroimaging research and their applicability to individual patients [21]. Studies targeting individual-level analyses are still limited as they require a large cohort of healthy controls (HC) scanned with the same MRI equipment and protocol, as well as advanced image post-processing techniques, which currently restricts their use in clinical practice [22, 23]. A recent study using multimodal MRI layering to model primary motor cortex pathology has identified distinct profiles associated with patient's rate of disease progression [24]. This study demonstrated the potential value of advanced MRI techniques to capture the in vivo heterogeneity of pathology maps, reflecting the specific clinical involvement across each of the 12 patients studied.

Additional MRI techniques that probe energy metabolism such as sodium ($^{23}$Na) MRI reported sodium homeostasis disturbances in the corticospinal tracts (CST) of ALS patients [25–27], reflecting metabolic alterations and energetic failure contributing to the neurodegenerative process [28].

The present study aims at studying disease severity at the individual level in ALS patients by mapping abnormal sodium homeostasis with brain $^{23}$Na MRI using a novel approach designed for individual patient.

## 2. Methods

### 2.1 Participants

This prospective study was approved by the local Ethics Committee (Comité de Protection des Personnes Sud Méditerranée 1), and written informed consent was obtained from all participants before the study began, in accordance with the tenets of the Declaration of Helsinki. Twenty-eight patients with ALS were consecutively recruited from the ALS reference center of our university hospital from 23 September 2015 to 30 August 2017. The inclusion criteria were a diagnosis of ALS according to the revised El Escorial criteria, no current or past history of neurologic disease other than ALS, no frontotemporal dementia, respiratory insufficiency, or substantial bulbar impairment incompatible with an MRI exam. Sixty-two HC with no history of neurologic or neuropsychiatric disorder were included. Their images, after verification of the absence of anomalies on anatomical MRI, were used to constitute the local database (n = 62, 36F/26M, mean age 40 ± 14 (sd) years-old; range [21; 66]).

### 2.2 Clinical exam

The clinical assessment of patients included onset site, disease duration, ALSFRS-R, ALSFRS-R slope corresponding to [48-ALSFRS-R]/disease duration in months [29–31], and neurological exam. All patients were followed up in the ALS center which allowed for collection of the ALSFRS-R 6 months after the MRI and the date of death.

### 2.3 MRI acquisition

MRI acquisition was performed on a 3T Verio system (Siemens, Erlangen, Germany) using a 32-channel phased-array [1]H head coil (Siemens, Erlangen, Germany) and a [23]Na-[1]H volume head coil (RAPID Biomedical, Rimpar, Germany). The sodium ([23]Na) MRI protocol included a 3D density-adapted radial sequence (TR/TE = 120/0.2ms; 17000 projections with 369 samples per projection; isotropic voxel size of 3.6mm; acquisition time = 34 min). Two tubes (50 mmol/L within 2% of agar gel) placed within the FOV served as a reference for quantification [25, 32].

The [1]H MRI protocol included a 3D T1-weighted (T1w) Magnetization-Prepared Rapid Acquisition Gradient-Echo (MPRAGE) sequence (TE/TR/TI = 3/2300/900ms, 160 slices, isotropic voxel size of 1mm, acquisition time = 6 min).

### 2.4 Image processing

Three-dimensional sodium images were reconstructed offline, denoised and normalized relative to signal from reference tubes to obtain quantitative TSC maps of the whole brain as detailed in a previous study [25] and illustrated in Fig 1.

The three-dimensional [23]Na and [1]H images were coregistered without resectioning. The [1]H images (MPRAGE) were normalized into the Montreal Neurologic Institute (MNI) template, and the resulting transformation was applied to the quantitative [23]Na maps. Finally, the obtained normalized quantitative TSC maps were smoothed using an 8-mm full width at half maximum Gaussian kernel [25, 32].

For measure of atrophy, bias field correction N4 was used to remove [1]H images intensity inhomogeneities [33]. [1]H images were classified into tissue types (grey matter (GM), white matter (WM) and cerebrospinal fluid (CSF)) using the Computational Anatomy toolbox (CAT12) [34]. Brain volumes were normalized for head size using the intracranial volume and GM fraction and WM fraction were computed. To map atrophy, normalized and

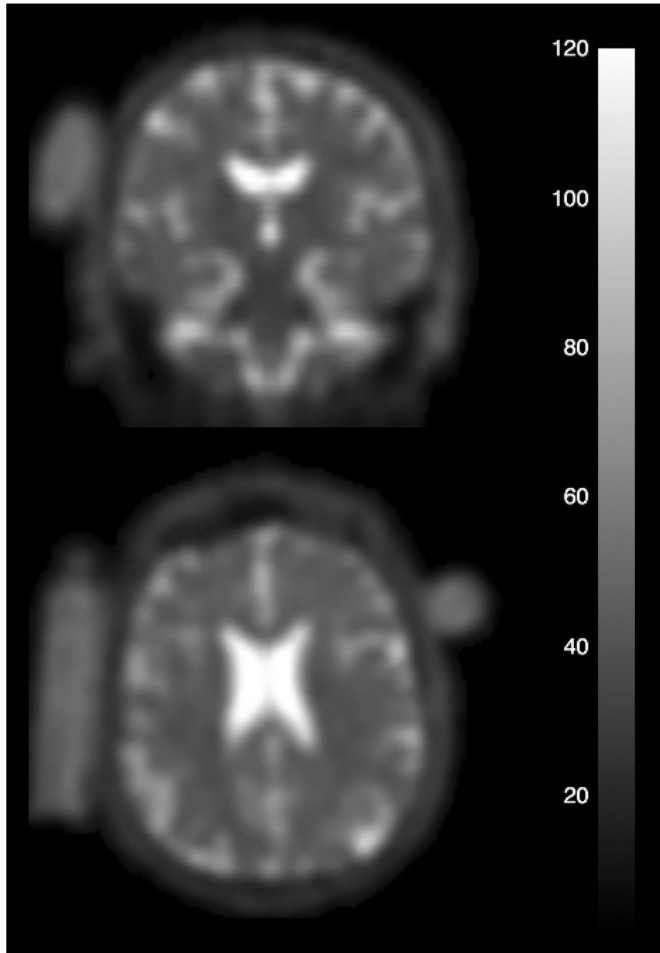

**Fig 1. Example of a 3D quantitative total sodium concentration map.**

smoothed GM fraction map of each patient was compared to maps of the whole group of HC by voxel-based statistical mapping analysis (SPM12, p = 0.005, false discovery rate corrected).

## 2.5 Statistical analyses

To map sodium homeostasis disturbances for each patient, we performed a voxel-based statistical mapping analysis (SPM12) on the normalized and smoothed TSC map for each patient compared to the control population to evidence voxels with TSC increase at the individual level as detailed in [35]. The statistical threshold of significant TSC increase was determined as the maximum p-value for which no significant cluster survived when comparing each control to the whole control population, as proposed in [36]. This conservative criterion allows for minimizing the false-positive cluster potentially observed in patients. Finally, for each patient, we determined the percentage of voxels with TSC increase, defined as the ratio of the number of voxels with TSC increase to the total number of voxels comprising the MNI brain template (n = 603220). This allowed to obtain a normalization rate of TSC increase for each subject, which facilitates comparisons between patients. This normalization rate has been referred in this manuscript as 'the percentage of voxels with TSC increase' which corresponds to the biomarker assessing abnormal sodium homeostasis.

To explore potential links between TSC increase and clinical parameters, patients have been splitted into two groups depending on the median of percentage of voxels with TSC increase for all patients. Half patients were considered as the group with low percentage of TSC increase while the second half constituted the group with high percentage of TSC increase. Between-group comparisons (p<0.05) were computed using the nonparametric Kruskal-Wallis test for the ALSFRS-R, the ALSFRS-R slope, the ALSFRS at 6 months and the survival using JMP Pro 16 software (SAS Institute, Cary, NC).

## 3. Results

### 3.1 Study participants

Clinical characteristics of the 28 ALS patients are reported in Table 1.

### 3.2 Statistical threshold

Individual TSC analyses comparing each HC to the whole group of HC allow for determining a maximum p value with no significant cluster when comparing any subject to the whole population of HC, p = 0.001, false discovery rate corrected.

### 3.3 Individual maps of TSC increase in patients

Individual maps quantifying voxels with TSC increase evidenced a high heterogeneity between patients in terms of percentage of voxels and spatial distribution. Patients' distribution according to the percentage of voxels with TSC increase within the whole brain is represented in Fig 2. Among the 28 patients, half showed TSC increase within more than 1.16% of whole brain voxels and 6 patients had no TSC increase. These latter 6 patients had a slow ALSFRS-R slope (<-0.3/month) (3 patients), and/or a lower motor neuron predominant ALS (5 patients), and/or a short disease duration (<8 months) (3 patients).

Individual maps showed that TSC increase was predominantly located in CSTs and fronto-temporal areas, but also evidenced a high heterogeneity between patients in terms of

**Table 1. Clinical characteristics of ALS patients.**

| Variable | ALS patients (n = 28) |
|---|---|
| Sex M/F | 19/9 |
| Age (y) | 54 ± 10 [35–70] |
| Disease duration (months) | 18.3 ± 14.5 [5–61] |
| Disease onset site | Bulbar 7 |
| | Upper limb 6 |
| | Lower limb 15 |
| Revised El Escorial diagnostic criteria | Definite 7 |
| | Probable 11 |
| | Probable laboratory supported 4 |
| | Possible 6 |
| ALSFRS-R | 39.0 ± 5.7 [23–47] |
| ALSFRS-R slope (/month) | -0.84 ± 0.88 [-0.12--3.57] |
| Survival from disease onset (months) | 56.1 ± 34.6 [10–128] |

Data are expressed as mean ± standard deviation and data between brackets represent the range [min-max].

ALSFRS-R: revised amyotrophic lateral sclerosis functional rating scale

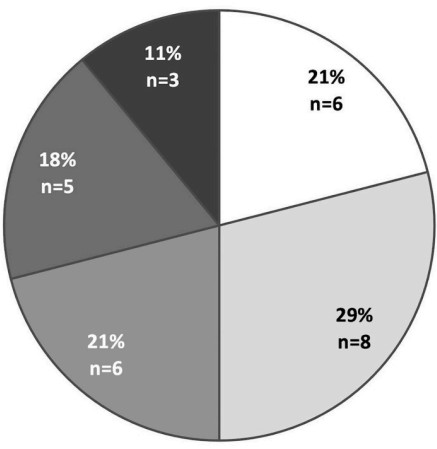

% of TSC increase  ☐ 0%  ☐ ]0% - 1.16%]  ▨ ]1.16% - 5%]  ▩ ]5% - 10%]  ■ >10%

**Fig 2. Patients distribution according to percentage of total sodium concentration (TSC) increase within the whole brain.**

percentage of voxels and spatial distribution that were overall consistent with the patient clinical condition, as illustrated in Fig 3.

Patient A was a 50-year-old woman with sporadic ALS whose disease duration was 13 months and who presented with a deficit limited to the lower limbs at the time of the MRI, predominantly on the left side, walking with a cane. ALSFRS-R slope was -0.54/month. TSC increase was 0.43% of voxels and located in the superior part of the CSTs, predominantly in the right side. Patient B was a 41-year-old man with sporadic, rapidly progressive ALS (ALSFRS-R slope -1.75/month), disease duration at the time of the MRI was 12 months. At the time of the MRI, the patient presented with spastic tetraparesis, used a wheelchair, needed feeding assistance, and had bulbar involvement with dysarthria and occasional choking. TSC increase was 5.32% of voxels and located in CSTs including bulbar areas. Patient C was a 43-year-old sporadic ALS male patient with right upper limb onset 11 months previously and presenting, at the time of the MRI, a deficit in both upper limbs, without deficit in the lower limbs or bulbar involvement. ALSFRS slope was -0.82/month. TSC increase was 3.75% of total voxels and located also in CSTs but without involving the brainstem. Patient D was a 59-year-old man with familial ALS linked to *C9orf72* repeat expansion, presenting at the time of the MRI with bulbar involvement (severe dysarthria, dietary consistency changes) without weakness in the limbs or cognitive impairment (Edinburgh Cognitive and Behavioural ALS Screen total score was 100 and ALS specific score was 87, years of education <12). Disease duration at the time of the MRI was 11 months and ALSFRS-R slope -0.72/month. TSC increase was high (16.56% of voxels) and located in frontotemporal areas, where atrophy was also present. Patient E was a 56-year-old man with sporadic ALS presenting with bulbar onset, very slow ALSFRS-R slope (-0.21/month) and very long disease duration at the time of the MRI (52 months). At the time of the MRI, he had bulbar involvement (intelligible spastic dysarthria) and spastic paraparesis (walking with a cane), and no other neurological sign especially no cerebellar syndrome. TSC increase was important (14.97% of voxels), located in CSTs and widespread in the brainstem and the cerebellum. Patient F was a 40-year-old male sporadic ALS patient with right upper limb onset 9 months previously and presenting, at the time of the MRI, a slight deficit in both upper limbs, without deficit in the lower limbs or bulbar involvement. ALSFRS-R slope was very slow (-0.12/month). No TSC increase was detected (0%).

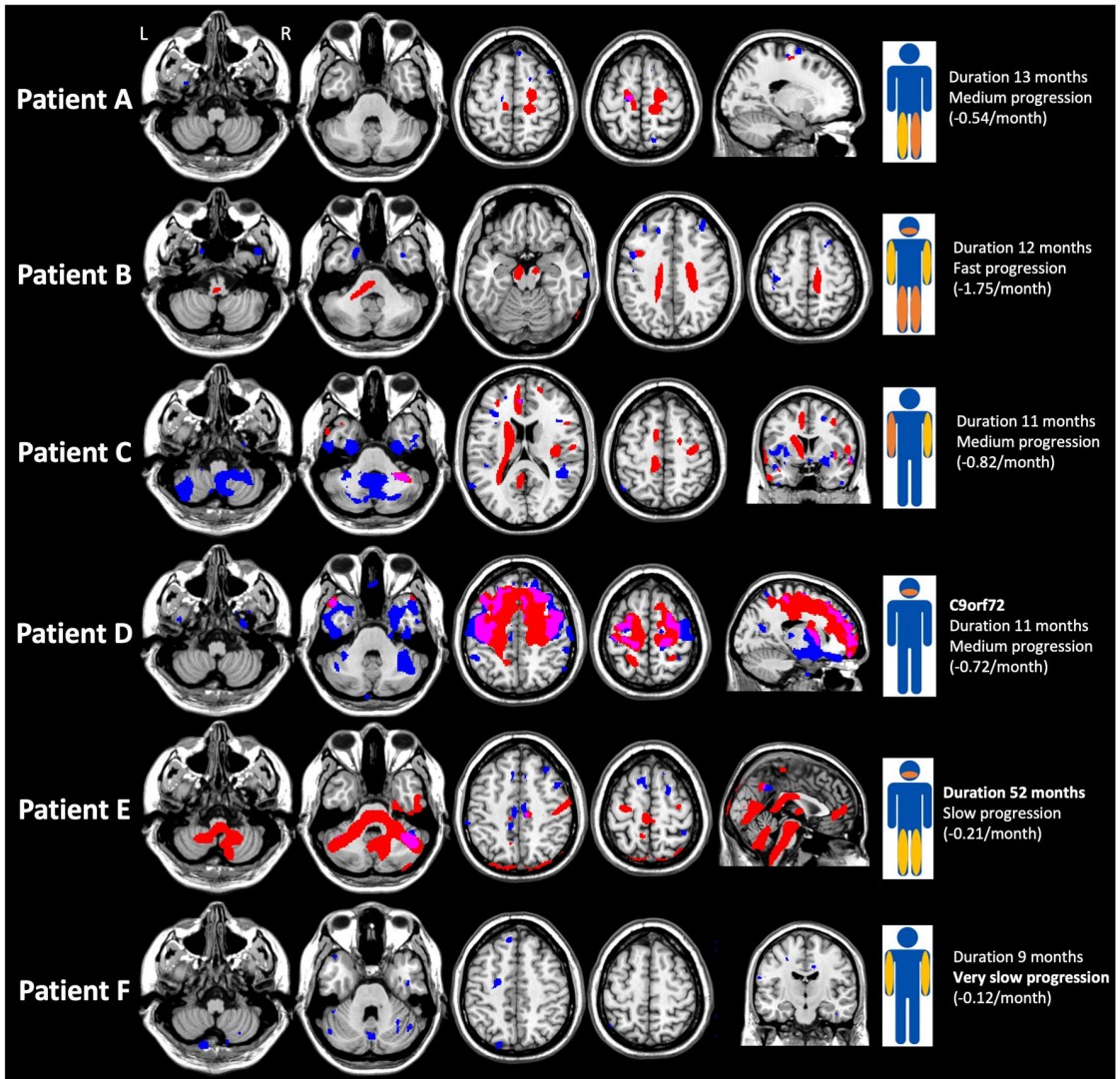

**Fig 3. Individual maps of sodium homeostasis disturbances.** For each individual map, voxels with total sodium concentration (TSC) increase are represented in red and atrophy in blue. The overlap between TSC increase and atrophy is represented in pink. The patient's clinical presentation is schematized by the level of the deficit (bulbar, right or left upper limb or lower limb) and its intensity (yellow for mild and orange for severe).

Except for Patient D who was *C9orf72* positive, genetic tests of the five other patients (*C9orf72* and *SOD1*) were negative. The individual maps of TSC increase of the other patients (except those with no TSC increase) are showed in S1 Fig.

Anonymized clinical and MRI data for each patient are provided in S1 Raw data.

### 3.4 Association between TSC increase and clinical data

Increase TSC was associated with a worse clinical disease evolution: patients with increase TSC ($>1.16\%$) showed higher ALSFRS-R slope ($-1.08 \pm 0.83$ vs $-0.59 \pm 0.89$, $p = 0.02$), lower ALSFRS-R at 6 months ($29.5 \pm 10.0$ vs $36.9 \pm 7.4$, $p = 0.04$) and shorter survival ($42 \pm 36$ vs $70 \pm 28$ months, $p = 0.04$), but no significant difference in ALSFRS-R at the time of the MRI

(37.4 ± 5.1 vs 40.1 ± 6.0, p = 0.13) nor in disease duration (16 ± 14 vs 21± 15, p = 0.23). No difference in GM atrophy (p = 0.34) neither WM atrophy (p = 0.96) was found.

## 4. Discussion

This pilot study evidenced sodium homeostasis disturbances at the individual level in ALS patients through [23]Na MRI. While the first study of [23]Na MRI in ALS [25] showed a common pattern of sodium abnormality in the group of ALS patients compared to healthy subjects, localized within the corticospinal tracts and the corpus callosum, the present study analyzing individual TSC map highlighted a large heterogeneity between patients, consistent with clinical presentation and severity.

While exploring disease activity at the level of the individual patient is needed in clinical practice to characterize the heterogeneity of the disease and help stratifying patients, only few individual MRI studies have been performed so far in ALS [22, 37] considering the important drawbacks and methodological challenges [38]. Interestingly, in the present study, increase TSC assessed by [23]Na MRI was associated with the clinical evolution of patients (ALSFRS-R slope, ALSFRS-R at 6 months, and survival). In ALS, the evolution of the disease is variable between patients, therefore biomarkers associated with disease severity and progression are needed at the individual level. However, associations between brain imaging metrics and clinical data are often difficult to establish in ALS because of the clinical heterogeneity of patients [39, 40]. Besides, motor disability in ALS is not only due to upper but also lower motor neuron degeneration which is not assessed by brain MRI. Indeed, in the present study, among the 6 patients with no TSC increase, 5 had clinically lower motor neuron predominant ALS. The association between TSC increase and ALSFRS-R at 6 months and survival found in this study suggests that [23]Na MRI could be a marker of individual disease prognosis.

Individual maps showed that TSC increase predominates in the CST as found in the previous study at the group level [25]. Furthermore, the topography and extent of the TSC increase seems to be consistent with the clinical presentation of the patients. Indeed, TSC was increased in the brainstem in patients with bulbar involvement. In Fig 3, Patient E who showed the most widespread TSC increase in the brainstem had the particularity of having a bulbar involvement evolving for a very long time. Patients showing no TSC increase had slow progression of disease disability and/or a lower motor neuron predominant ALS and/or a short disease duration. Particularly noteworthy, the patient with *C9orf72* familial ALS (Patient D) had diffuse TSC increase within frontotemporal regions, whereas clinically he had only bulbar involvement without limb weakness nor cognitive impairment at the time of the MRI. The *C9orf72* expansion, causing ALS, frontotemporal dementia (FTD) or ALS-FTD phenotypes, is the most common mutation in familial (40%) and sporadic (5–10%) ALS [41]. This result is in agreement with data from other MRI techniques reporting a different signature in *C9orf72* patients compared to other ALS patients, including extensive, relatively symmetric volume loss and cortical thinning, and diffusion changes in frontal areas occurring early in disease, even in presymptomatic *C9orf72* mutation carriers [42–45].

TSC increase measured by [23]Na MRI evidenced sodium homeostasis disturbances in brain tissue. While the pathophysiological processes inducing these sodium alterations are not well known, the hypotheses to explain these increase sodium concentrations in neurodegenerative diseases are related to mitochondrial dysfunction and subsequent energy failure leading to neuronal degeneration [46]. TSC increase was also reported in other neurodegenerative conditions such as Parkinson disease [47], Huntington disease [48] and in progressive multiple sclerosis [49, 50].

The present pilot study suffers from several limitations. Further studies including more ALS patients are needed to confirm our results, better characterize the heterogeneity of individual TSC maps between patients and analyze the relationship between patient's clinical presentation or genetic status and TSC increase. In addition, a normative database with a higher number of subjects would improve the sensitivity of the technique. Furthermore, longitudinal [23]Na MRI study in ALS would also benefit for exploring the potential link between sodium homeostasis changes over time and the clinical prognosis of each patient. Nevertheless, longitudinal MRI studies are difficult in ALS because most patients have a rapid clinical worsening including respiratory and bulbar involvement, which quickly makes the MRI examination impossible. Future research using advanced techniques at 7T MRI, would allow assessing not only the total sodium signal, which reflects the average sodium content from both intracellular and extracellular compartments, but also focusing specifically on intracellular sodium [31, 32]. This would lead to a specific exploration of energy metabolism through sodium homeostasis within the structural brain networks of ALS patients. This approach would provide valuable insights into the processes of brain reorganization and deepen our understanding of the mechanisms underlying neurodegeneration.

In conclusion, this pilot study allowed to map abnormal sodium homeostasis with brain [23]Na MRI in ALS using a novel approach designed for individual patient. Individual maps of increase TSC showed heterogeneity between patients and an association with disease presentation and severity. Mapping sodium homeostasis disturbances at the individual level in ALS patients might therefore be a future biomarker to help stratify patients and evaluate new therapeutics.

## Supporting information

**S1 Fig. Additional individual maps of sodium homeostasis disturbances of patients.** For each individual map, voxels with total sodium concentration (TSC) increase are represented in red and atrophy in blue. TSC increase can be related to atrophy in some patients as shown by the overlap between TSC increase and atrophy represented in pink.
(TIF)

**S1 Raw data. Anonymized clinical and MRI data for each patient.**
(XLS)

## Acknowledgments

We thank Lauriane Pini, Patrick Viout, Claire Costes, Véronique Gimenez, and Hafida Kribich for their help on data acquisition.

## Author Contributions

**Conceptualization:** Aude-Marie Grapperon, Jean-Philippe Ranjeva, Maxime Guye, Annie Verschueren, Shahram Attarian, Wafaa Zaaraoui.

**Data curation:** Aude-Marie Grapperon, Mohamed Mounir El Mendili, Jean-Philippe Ranjeva, Wafaa Zaaraoui.

**Formal analysis:** Aude-Marie Grapperon, Mohamed Mounir El Mendili, Wafaa Zaaraoui.

**Funding acquisition:** Aude-Marie Grapperon, Jean-Philippe Ranjeva, Maxime Guye, Shahram Attarian, Wafaa Zaaraoui.

**Investigation:** Aude-Marie Grapperon, Adil Maarouf, Maxime Guye, Annie Verschueren, Shahram Attarian, Wafaa Zaaraoui.

**Methodology:** Aude-Marie Grapperon, Mohamed Mounir El Mendili, Adil Maarouf, Jean-Philippe Ranjeva, Maxime Guye, Wafaa Zaaraoui.

**Project administration:** Aude-Marie Grapperon, Jean-Philippe Ranjeva, Maxime Guye, Annie Verschueren, Shahram Attarian, Wafaa Zaaraoui.

**Resources:** Aude-Marie Grapperon, Maxime Guye, Annie Verschueren, Shahram Attarian, Wafaa Zaaraoui.

**Software:** Aude-Marie Grapperon, Mohamed Mounir El Mendili, Jean-Philippe Ranjeva, Maxime Guye, Wafaa Zaaraoui.

**Supervision:** Aude-Marie Grapperon, Mohamed Mounir El Mendili, Jean-Philippe Ranjeva, Maxime Guye, Shahram Attarian, Wafaa Zaaraoui.

**Validation:** Aude-Marie Grapperon, Jean-Philippe Ranjeva, Maxime Guye, Annie Verschueren, Shahram Attarian, Wafaa Zaaraoui.

**Visualization:** Aude-Marie Grapperon, Jean-Philippe Ranjeva, Maxime Guye, Annie Verschueren, Shahram Attarian, Wafaa Zaaraoui.

**Writing – original draft:** Aude-Marie Grapperon, Jean-Philippe Ranjeva, Maxime Guye, Shahram Attarian, Wafaa Zaaraoui.

**Writing – review & editing:** Aude-Marie Grapperon, Jean-Philippe Ranjeva, Maxime Guye, Shahram Attarian, Wafaa Zaaraoui.

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
