## [Decision Letter · Decision Letter 0]

23 Sep 2024

PONE-D-24-27226In vivo mapping of sodium homeostasis disturbances in individual ALS patients: a brain 23Na MRI studyPLOS ONE

Dear Dr. Grapperon,

Thank you for submitting your manuscript to PLOS ONE. After careful consideration, we feel that it has merit but does not fully meet PLOS ONE’s publication criteria as it currently stands. Therefore, we invite you to submit a revised version of the manuscript that addresses the points raised during the review process.

We look forward to receiving your revised manuscript.

Kind regards,

Belgin Sever, Ph.D.

Academic Editor

PLOS ONE

**Journal Requirements:**

- https://www.ismrm.org/24/docs/24ToC.pdf

In your revision ensure you cite all your sources (including your own works), and quote or rephrase any duplicated text outside the methods section. Further consideration is dependent on these concerns being addressed.

This research was funded by Assistance Publique Hopitaux de Marseille (APHM, AORC Junior program), ARSLA (Association pour la Recherche sur la Sclérose Latérale Amyotrophique et autres maladies du motoneurone) and FRC (Fédération pour la Recherche sur le Cerveau).

5. We notice that your supplementary figures are uploaded with the file type 'Figure'. Please amend the file type to 'Supporting Information'. Please ensure that each Supporting Information file has a legend listed in the manuscript after the references list.

Reviewers' comments:

Reviewer's Responses to Questions

**Comments to the Author**

1. Is the manuscript technically sound, and do the data support the conclusions?

Reviewer #1: Yes

2. Has the statistical analysis been performed appropriately and rigorously? 

Reviewer #1: Yes

3. Have the authors made all data underlying the findings in their manuscript fully available?

Reviewer #1: No

4. Is the manuscript presented in an intelligible fashion and written in standard English?

Reviewer #1: Yes

5. Review Comments to the Author

**Reviewer #1:** Review of Paper PONE-D-24-27226

Title:

In vivo mapping of sodium homeostasis disturbances in individual ALS patients: a brain 23Na MRI study

Summary:

The purpose of this study was to investigate the severity of ALS at the individual level using brain 23Na MRIs in ALS patients. The study used (_ ^23)Na MRI to map sodium homeostasis disturbances in ALS patients, revealing that the increase in total sodium concentration (TSC) varies among individuals. This variation is linked to clinical presentation and disease severity. The findings suggest that TSC increase could serve as a useful marker for the clinical heterogeneity in ALS, which could significantly impact the outcomes of therapeutic trials.

Strengths And Weaknesses:

Strengths

It has a thorough explanation of the main idea.

Clearly explain the methodology.

As a case study it has a good experimental design.

Weaknesses

Overall, the writing can be significantly improved to address the following concerns.

Introduction: You can add more references in the related work to show that you check new works.

Methodology:

Put Figures in the right place.

You didn’t define a clear biomarker based on your finding

Normalize rate for each subject

Questions:

Answering the following questions and addressing the weaknesses above can significantly improve my score.

How can you include the effect of normalization rate foe each subject?

What is your suggestion for future work?

6. PLOS authors have the option to publish the peer review history of their article (what does this mean?). If published, this will include your full peer review and any attached files.

Reviewer #1: No

---

## [Author Response · Author response to Decision Letter 0]

26 Nov 2024

Dear Editor and Reviewer,

We thank you for taking the time and effort to provide insightful guidance toward improving our manuscript. A detailed response to each point raised by you is given below in the order in which they appeared in the letter. Changes to the manuscript have been highlighted using red colored text.

1. Reviewer’s comments

Weaknesses

Overall, the writing can be significantly improved to address the following concerns.

● Introduction: You can add more references in the related work to show that you check new works.

=> Response: As suggested, we added several recent references in the Introduction.

The following references have been added:

1) Recent works reporting widespread structural damage in motor but also non-motor brain regions in ALS patients:

- McMackin R, et al. Biomarkers in amyotrophic lateral sclerosis: current status and future prospects. Nat Rev Neurol. 2023

- Rajagopalan V et Pioro EP. Graph theory network analysis reveals widespread white matter damage in brains of patients with classic ALS. Amyotroph Lateral Scler Frontotemporal Degener. 2024

- Wendebourg MJ, et al. The Lateral Corticospinal Tract Sign: An MRI Marker for Amyotrophic Lateral Sclerosis. Radiology. 2024

- Tahedl M, et al. Progressive Cerebrocerebellar Uncoupling in Sporadic and Genetic Forms of Amyotrophic Lateral Sclerosis. Neurology. 2024

- Müller HP, et al. Temporal and spatial progression of microstructural cerebral degeneration in ALS: A multicentre longitudinal diffusion tensor imaging study. Neuroimage Clin. 2024.

- Spinelli EG, et al. Structural and Functional Brain Network Connectivity at Different King's Stages in Patients With Amyotrophic Lateral Sclerosis. Neurology. 2024

- Basaia S, et al. Structural and functional brain connectome in motor neuron diseases: A multicenter MRI study. Neurology. 2020

- Pandya S, et al. Modeling seeding and neuroanatomic spread of pathology in amyotrophic lateral sclerosis. Neuroimage. 2022

- Tilsley P, et al. Neurofilament Light Chain Levels Interact with Neurodegenerative Patterns and Motor Neuron Dysfunction in Amyotrophic Lateral Sclerosis. AJNR Am J Neuroradiol. 2024

2) Recent neuroimaging works reporting different patterns of the disease, illustrating the heterogeneity of ALS:

- Tan HHG, et al. MRI Clustering Reveals Three ALS Subtypes With Unique Neurodegeneration Patterns. Ann Neurol. 2022 

- Bede P, et al. Clusters of anatomical disease-burden patterns in ALS: a data-driven approach confirms radiological subtypes. J Neurol. 2022 

- Feng F, et al. Different patterns of structural network impairments in two amyotrophic lateral sclerosis subtypes driven by 18F-fluorodeoxyglucose positron emission tomography/magnetic resonance hybrid imaging. Brain Commun. 2024 

- Milella G, et al. Clinical Profiles and Patterns of Neurodegeneration in Amyotrophic Lateral Sclerosis: A Cluster-Based Approach Based on MR Imaging Metrics. AJNR Am J Neuroradiol. 2023 

3) A recent review of neuroimaging in ALS:

- Tu S, et al. Pathological insights derived from neuroimaging in amyotrophic lateral sclerosis: emerging clinical applications. Curr Opin Neurol. 2024

4) Recent neuroimaging studies providing information on disease progression at individual level:

- Northall A, et al. Multimodal layer modelling reveals in vivo pathology in amyotrophic lateral sclerosis. Brain. 2024

- Meier JM, et al. Connectome-Based Propagation Model in Amyotrophic Lateral Sclerosis. Ann Neurol. 2020 

5) We have also included references to recent neuroimaging studies in C9orf72 ALS in the discussion section:

- Wiesenfarth M, et al. Structural and microstructural neuroimaging signature of C9orf72-associated ALS: A multiparametric MRI study. Neuroimage Clin. 2023

- Bede P, et al. Presymptomatic grey matter alterations in ALS kindreds: a computational neuroimaging study of asymptomatic C9orf72 and SOD1 mutation carriers. J Neurol. 2023 

- Nigri A, et al. C9orf72 ALS mutation carriers show extensive cortical and subcortical damage compared to matched wild-type ALS patients. Neuroimage Clin. 2023

The introduction has been modified to include these references:

‘Several MRI techniques, such as diffusion tensor imaging (DTI) or cortical thickness measurements, have reported widespread structural damage in motor but also non-motor brain regions in ALS patients at the population level [4–16]. Brain damage revealed by MRI techniques in patients with ALS is well-documented but remains inconsistent due to the high heterogeneity of the disease. Recently, imaging techniques have helped to identify distinct disease-burden patterns, aiding in patient stratification [17–20]. However, the majority of studies focus on group comparisons, typically between ALS patients and healthy controls, which has led to a growing gap between group-level findings in ALS neuroimaging research and their applicability to individual patients [21]. Studies targeting individual-level analyses are still limited as they require a large cohort of healthy controls (HC) scanned with the same MRI equipment and protocol, as well as advanced image post-processing techniques, which currently restricts their use in clinical practice [22,23]. A recent study using multimodal MRI layering to model primary motor cortex pathology has identified distinct profiles associated with patient's rate of disease progression [24]. This study demonstrated the potential value of advanced MRI techniques to capture the in vivo heterogeneity of pathology maps, reflecting the specific clinical involvement across each of the 12 patients studied.’

● Methodology: 

○ Put Figures in the right place.

=> Response: As recommended, we placed the Figures in the right place within the manuscript.

○ You didn’t define a clear biomarker based on your finding

=> Response: To clarify this point, we added the following sentence in the manuscript:

Methods section: ‘…This normalization rate has been referred in this manuscript as ‘the percentage of voxels with TSC increase’ which corresponds to the biomarker assessing abnormal sodium homeostasis.’

○ Normalize rate for each subject

Questions:

Answering the following questions and addressing the weaknesses above can significantly improve my score.

1. How can you include the effect of normalization rate foe each subject?

=> Response: To clarify this point, we added details in the ‘Methods’ section (Statistical Analysis sub-section) to emphasize that we used an approach to obtain a normalization rate for each subject allowing comparison between subjects:

Methods section: ‘for each patient, we determined the percentage of voxels with TSC increase, defined as the ratio of the number of voxels with TSC increase in the individual to the total number of voxels comprising the MNI brain template (n=603220). This allowed us to obtain a normalization rate of TSC increase for each subject, which facilitates comparisons between patients. This normalization rate has been referred in this manuscript as ‘the percentage of voxels with TSC increase’ which corresponds to the biomarker assessing abnormal sodium homeostasis.’

2. What is your suggestion for future works?

=> Response: As suggested, we added in the ‘Discussion’ section several perspectives that we intent to further investigate.

Discussion section: ‘Future research using advanced techniques at 7T MRI, would allow assessing not only the total sodium signal, which reflects the average sodium content from both intracellular and extracellular compartments, but also focusing specifically on intracellular sodium [31,32]. This would lead to a specific exploration of energy metabolism through sodium homeostasis within the structural brain networks of ALS patients. This approach would provide valuable insights into the processes of brain reorganization and deepen our understanding of the mechanisms underlying neurodegeneration.’

2. Editor’s comments

Journal Requirements: 

=> Response: This point has been checked carefully.

https://www.ismrm.org/24/docs/24ToC.pdf

In your revision ensure you cite all your sources (including your own works), and quote or rephrase any duplicated text outside the methods section. Further consideration is dependent on these concerns being addressed. 

=> Response: We apologize for this minor occurrence of overlapping text which came from a short abstract with preliminary data of this work that was selected for an oral communication at the 2024 ISMRM conference. As suggested, we added the reference of our own work in the introduction section ‘ref 28’.

This research was funded by Assistance Publique Hopitaux de Marseille (APHM, AORC Junior program), ARSLA (Association pour la Recherche sur la Sclérose Latérale Amyotrophique et autres maladies du motoneurone) and FRC (Fédération pour la Recherche sur le Cerveau). 

=> Response: ‘The funders had no role in study design, data collection and analysis, decision to publish, or preparation of the manuscript.’

=> Response: All relevant data are within the manuscript and its supporting information files. Due to legal and ethical constraints surrounding potentially identifying information, MRI data cannot be shared publicly. Requests for access to the data will be evaluated on a case-by-case basis, subject to a data usage agreement (please email aude-marie.grapperon@ap-hm.fr).

5. We notice that your supplementary figures are uploaded with the file type 'Figure'. Please amend the file type to 'Supporting Information'. Please ensure that each Supporting Information file has a legend listed in the manuscript after the references list.

=> Response: This issue was corrected.

=> Response: As suggested by the reviewer, we added recent references in the ‘Introduction’ section. No retracted article was cited.

---

## [Editor Report · Decision Letter 1]

19 Dec 2024

In vivo mapping of sodium homeostasis disturbances in individual ALS patients: a brain 23Na MRI study

PONE-D-24-27226R1

Dear Dr. Grapperon,

We’re pleased to inform you that your manuscript has been judged scientifically suitable for publication and will be formally accepted for publication once it meets all outstanding technical requirements.

Kind regards,

Belgin Sever, Ph.D.

Academic Editor

PLOS ONE
---

## [Editor Report · Acceptance letter]

11 Jan 2025

PONE-D-24-27226R1 

PLOS ONE

Dear Dr. Grapperon, 

I'm pleased to inform you that your manuscript has been deemed suitable for publication in PLOS ONE. Congratulations! Your manuscript is now being handed over to our production team.

Kind regards, 

on behalf of

Assoc. Prof. Dr. Belgin Sever 

Academic Editor

PLOS ONE